# Movement ecology of captive-bred axolotls in restored and artificial wetlands: Conservation insights for amphibian reintroductions and translocations

Alejandra G. Ramos[1,2], Horacio Mena[2], David Schneider(ID)[1,3], Luis Zambrano(ID)[2*]

1 Facultad de Ciencias, Universidad Autónoma de Baja California, Ensenada, Baja California, México, 2 Departamento de Zoología, Instituto de Biología, Universidad Nacional Autónoma de México, Ciudad de México, México, 3 Departamento de Biología de la Conservación, Centro de Investigación Científica y de Educación Superior de Ensenada, Ensenada, Baja California, México

* zambrano@ib.unam.mx

## Abstract

Amphibians are among the most endangered vertebrates globally due to habitat loss, environmental degradation, and urban expansion. The axolotl (*Ambystoma mexicanum*), a critically endangered aquatic species endemic to Lake Xochimilco, exemplifies these challenges. This study evaluates the viability of restored and artificial wetlands for axolotl conservation by comparing movement patterns, home range sizes, and habitat use. Using VHF telemetry, we tracked captive-bred axolotls released into both environments. Axolotls survived and foraged successfully in both sites, with those in an artificial pond in La Cantera Oriente exhibiting larger home ranges (mean: 2,747 m²) and greater daily distances traveled than those in a restored chinampa in Lake Xochimilco, where home ranges were smaller (mean: 382 m²). A quadratic relationship between water temperature and movement indicated a narrow thermal preference, with axolotl movement peaking at around 16–17°C in Xochimilco and 15.5–16.5°C in La Cantera Oriente, declining beyond these ranges. Additionally, in La Cantera Oriente, female axolotls traveled significantly greater daily distances than males, with females averaging 86.75 meters per day compared to 54.33 meters for males. In Xochimilco, daily distance traveled decreased with age. Recaptured individuals gained weight, suggesting successful adaptation, although two axolotls were lost to avian predation in Xochimilco after the study concluded. These findings highlight the potential of artificial wetlands like La Cantera Oriente for axolotl conservation by providing stable conditions that may mitigate habitat degradation and climate change impacts. The study recommends integrating native and artificial habitats into conservation strategies, incorporating predator awareness training before release, and ongoing habitat monitoring to enhance survival outcomes for this iconic species.

**Data availability statement:** Data supporting our study's findings are now hosted on Figshare and in supplementary materials. During the peer review process, the dataset can be accessed via a private link: https://figshare.com/s/5aae1c742de87b2ad75f. Upon acceptance of the manuscript, the data will be made publicly available at https://doi.org/10.6084/m9.figshare.27936111 (inactive during the peer review process).

**Funding:** This project was funded by UNAM PAPIIT No. 705 IV200117 and IV210117 Programa de Apoyo a Proyectos de Investigación e Innovación Tecnológica (PAPIIT-IV200117) AGR received a postdoctoral research grant from PAPIIT IV200117 and IV210117. The funders had no role in study design, data collection and analysis, decision to publish, or preparation of the manuscript.

**Competing interests:** The authors have declared that no competing interests exist.

## Introduction

Amphibians are the most threatened group of vertebrates worldwide, with 41% of species currently listed as at risk of extinction on the International Union for Conservation of Nature (IUCN) Red List [1,2]. In 2019, approximately 25% of amphibian species were classified as Data Deficient, with estimates suggesting that at least half of these species were also likely threatened [3]. Amphibians play a crucial role in sustaining ecosystem dynamics as both predators and prey in wetlands and serve as essential indicators of biodiversity and environmental health [4–6].Habitat loss, fragmentation, and degradation—including water pollution—are major drivers of global amphibian population declines and extinctions [7–10]. Urban expansion severely transforms freshwater ecosystems [5,11], disproportionately threatening amphibian species that are entirely dependent on aquatic habitats more than those that transition to terrestrial habitats as adults [12].

Among the amphibians most affected by these threats is the critically endangered axolotl (*Ambystoma mexicanum*) [2], a paedomorphic salamander that retains its juvenile features and aquatic lifestyle into adulthood [13]. *A. mexicanum* is one of 17 species of *Ambystoma* found in Mexico, of which 58.8% are considered globally threatened, and 94.1% are endemic to the country [14]. However, *A. mexicanum* is the species most commonly referred to as "axolotl" in scientific and conservation literature. Other notable Mexican *Ambystoma* species, *Ambystoma dumerilii* and *Ambystoma andersoni*, are locally called "achoque," a name from the Purépecha language that reflects their cultural significance in the Lake Pátzcuaro region of western Mexico [15]. In contrast, "axolotl" derives from Nahuatl, the language of the Aztec civilization that flourished in central Mexico. *A. mexicanum* is endemic to Lake Xochimilco, the last remnant of a once-extensive wetland system in the Central Valley of Mexico, where habitat loss has severely impacted its survival [16].

Lake Xochimilco has been managed for more than 1,500 years. During the Aztec Empire, wetlands were partially transformed into canals surrounding rectangular islands called *chinampas*, which supported intensive food production and promoted local biodiversity [17]. Today, however, Lake Xohimilco faces severe environmental challenges. Urban expansion, environmental degradation, invasive species, and a severe decline in water quality—driven by untreated wastewater discharge, agricultural runoff, and surrounding urbanization—have led to a marked decline in axolotl populations [18–21]. In response to these threats, the *chinampa-refuge* project was initiated in 2004 by the Laboratorio de Restauración Ecológica at the Universidad Nacional Autónoma de México (UNAM) in collaboration with local *chinampero* farmers. This project integrates traditional food production with aquatic conservation by creating refuges within *chinampas* to support native species, including the axolotl [17].However, persistent environmental pressures limit the effectiveness of conservation efforts, which remain largely confined to restored *chinampas*.

Given the intensifying amphibian extinction crisis, continuous and proactive human interventions, such as translocations, are now considered essential to preserve vulnerable species and prevent extinctions [22–25]. Translocation—the deliberate movement of living organisms from one area to another— is a key

conservation strategy that plays an active role in species recovery. It is often combined with captive breeding to restore dwindling or extirpated populations and to establish new populations in suitable habitats [26,27]. Although most translocations have involved birds and mammals [23,28], amphibian translocations are increasing in number and success [26,29]. However, programs for threatened salamanders remain scarce, primarily because few species are easily kept and bred in captivity [30]. Concerns about the fitness of captive-reared animals and their ability to adapt to wild conditions persist, with predation-induced mortality being a main cause of translocation failure [31]. Additionally, not addressing the initial causes of population decline before releasing organisms can hinder the success of these efforts [32].

The creation and restoration of wetlands have proven effective for the conservation of numerous amphibian species, including salamanders [33]. For instance, the establishment of over 200 ponds led to increases in both amphibian diversity and populations of the threatened crested newt (*Triturus cristatus*) [34]. Similarly, habitat restoration efforts are critical for species like the Mexican axolotl (*Ambystoma mexicanum*), whose natural habitat is wetlands such as Lake Xochimilco. Understanding the specific habitat preferences, movement behaviors, and survival rates of threatened species is essential for designing effective conservation programs because distinct microhabitats within an ecosystem can differentially affect species fitness [35]. Previous studies on amphibians like hellbenders and Chinese giant salamanders have shown that factors such as age, sex, and habitat characteristics play critical roles in determining movement patterns [36,37]. Nevertheless, relatively few programs have examined post-release survival and habitat use in detail [30]. Biotelemetry methods, such as very high frequency (VHF) radio telemetry, offer valuable tools for monitoring individuals, allowing researchers to assess movement ecology, habitat use, and factors influencing post-release success [38]. However, this method has limitations, including restricted signal range, limited battery life, potential malfunctions, susceptibility to adverse weather conditions, and high costs [39,40]. In small species like salamanders, which require transmitter implantation, the risk of mortality can be a significant concern [30,41,42]. However, this risk may be mitigated by extending post-surgery recovery time [43] and using smaller transmitters [42].

Conserving threatened species within their natural habitats remains the ideal strategy [35], as this approach is often more cost-effective and degraded habitats can still support biodiversity[44]. However, the ongoing challenges in Lake Xochimilco encourage the exploration of alternative habitats for axolotl conservation. Furthermore, as climate unpredictability is expected to worsen, it becomes even more crucial to identify and protect refuges that are anticipated to remain suitable over time [6]. La Cantera Oriente (LCO) is an example of how artificial wetlands can develop into biologically diverse habitats over time. Located within the protected area of the Reserva Ecológica del Pedregal de San Ángel (REPSA), LCO is an artificial wetland with four ponds and a stream, created approximately 30 years ago when basalt mining activities caused underground springs to surface. Over time, it has developed into a thriving habitat with diverse aquatic vegetation and native fauna, including potential prey species such as crayfish for adult axolotls [45] and zooplankton for larvae [46]. Additionally, LCO provides physicochemical conditions (e.g., temperature, pH, and dissolved oxygen) suitable for axolotl reproduction and survival, as demonstrated by successful egg-laying, hatching, and larval development into juveniles [46].

In this study, we employed very high frequency (VHF) telemetry to evaluate the movement behaviors, home range sizes, and habitat use of captive-bred axolotls released into both an artificial wetland (LCO) and a restored chinampa within their native habitat of Lake Xochimilco. By monitoring these individuals and analyzing factors influencing their movement and survival—such as environmental factors (temperature and time of day) and biological (sex, mass, and age) factors—we aim to assess the feasibility of both sites as viable locations for axolotl conservation. Our study offers practical implications for habitat restoration, reintroduction, and translocation programs, highlighting how both restored native habitats and artificial environments can play crucial roles in sustaining endangered species. Ultimately, this research aims to inform future conservation strategies by exploring how axolotls respond to different environmental conditions and emphasizing the importance of integrating both native and alternative habitats in their preservation.

## Materials and methods

### Study areas

This study was conducted in two distinct aquatic habitats located in southern Mexico City: a restored chinampa in Lake Xochimilco (hereafter Xochimilco) and a pond in the artificial wetland of La Cantera Oriente (LCO) (Fig 1). Aerial images of both study areas were captured using a DJI Mavic Pro Platinum drone (DJI, Shenzhen, China) equipped with a 1/2.3" CMOS sensor (12.35 MP effective pixels). For the LCO map, a single photograph was used, while the Xochimilco map was generated by stitching multiple overlapping images using Microsoft Image Composite Editor (ICE; Microsoft Corporation, Redmond, USA), the georeferencing was performed using MapTiler Engine 13.4 (MapTiler, Baar, Switzerland).

In Xochimilco, we used a 500 m² chinampa within the chinampa-refuge project, where canals are maintained without agrochemicals and equipped with filters to improve water quality and exclude exotic fish [47]. These refuges have proven

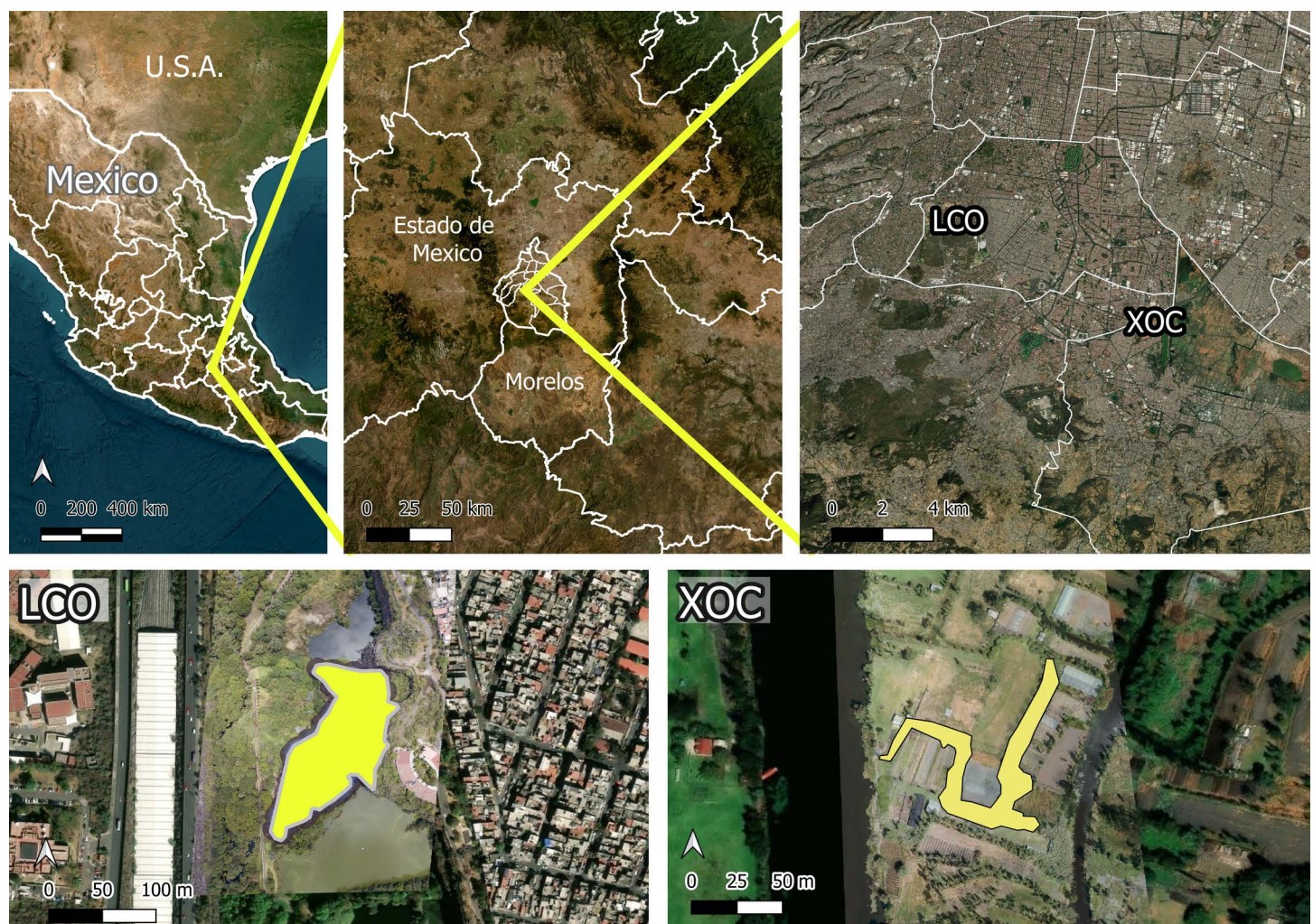

**Fig 1. Spatial location of study areas.** Overview of the two study areas in southern Mexico City: Xochimilco (XOC) and La Cantera Oriente (LCO). The top panels illustrate the geographic location of the study areas at three distinct spatial scales (Base map sources: Esri, Maxar, Earthstar Geographics, and the GIS User Community), while the bottom panels provide close-up views of each wetland (aerial photographs taken by David Schneider). Scale bars represent distances in kilometers (top) and meters (bottom) for reference.

beneficial for the survival and reproduction of axolotls and other native species [48]. They aim to recreate conditions similar to those in historical Lake Xochimilco, where axolotl populations once thrived.

In LCO, we selected a 7,059 m² pond previously identified as providing suitable conditions for axolotl reproduction and survival [46]. Furthermore, as part of a pilot study, we released a radio-tagged, captive-raised adult male and female pair into this pond. Both individuals survived and remained active over a five-week monitoring period. Upon recapture, we observed that the male had gained weight and showed no signs of health issues (unpublished data from authors). However, we were unable to recapture the female, as she did not enter our traps.

Summary statistics of physicochemical parameters recorded daily on monitoring days for both LCO and Xochimilco throughout the study, are provided in the supplementary materials (S1 Table).

### Ethics statement

All axolotls in this study were born and raised in captivity at the Laboratorio de Restauración Ecológica (LRE) of the Instituto de Biología, UNAM, under permit DGVS-PIMVS-CR-IN-1833-CDMX/17. This research complies with Mexican regulations and was conducted under the oversight of the Instituto de Biología's ethics committee, covering both fieldwork and colony care. No mortalities were observed as a result of transmitter implantation, and all axolotls survived the procedure and monitoring period.

### Study species and transmitters

For this study, we selected a total of 18 axolotls from the LRE colony, comprising nine females and nine males. Detailed information about the selected axolotls, including their age, mass, and length, is provided in S2 Table. The housing conditions in this colony, which have been consistently implemented since before the onset of this study and continue to the present, include maintaining axolotls without substrate or plants to minimize the risk of disease transmission. The axolotls are provided with artificial shelters that are routinely sterilized for their security and comfort. Their diet consists of live food such as Artemia, small fish, crayfish, and worms, all quarantined before feeding and offered every other day, complemented by monthly nutritional supplements. To prevent unintended breeding, opposite-sex axolotls are kept separately. An air-conditioning system is maintained to keep the water temperature below 18ºC, ensuring optimal health. Axolotls were implanted with SOPI-2038 transmitters from Wildlife Materials, Murphysboro, IL, USA, each programmed with a unique frequency. These transmitters, weighing an average of two g, constituted about 3% of the axolotls' body mass, well below the 12% limit deemed acceptable [49]. We used internal transmitters to minimize the risk of skin injuries and to avoid inhibiting natural behaviors such as the use of refugia, predator avoidance, and hunting, which can be affected by external transmitters [38,50,51].

The implantation process began with the axolotls being anesthetized in a 10-minute benzocaine bath (12 ml of benzocaine diluted in 1 L of water). Once sedated, they were weighed and measured. The surgical procedure was performed by Horacio Mena, a veterinarian specialized in axolotls and co-author of this paper. He made a 1-cm longitudinal ventral incision under sterile conditions, carefully inserted the transmitter into the coelomic cavity, and closed the skin with a pattern of spaced stitches, following the administration of analgesics and antibiotics [52]. After surgery, each axolotl was housed individually in a small tank and monitored daily throughout a two-week recovery period. Complete wound healing occurred within a few days without any signs of infection or inflammation.

### Data collection

We utilized a lightweight 3-element YAGI antenna coupled with a Wildlife Materials Inc. (Murphysboro, IL, USA) TRX-48S receiver for radio tracking. Monitoring at both sites was suspended during rainfall or thunderstorms to prevent equipment damage and ensure safety. Initially, we scanned for the unique frequencies of each individual to locate their transmitter

signals. Once an axolotl's signal was detected, team members rowed to the location in LCO or walked in Xochimilco, recording the exact decimal coordinates using a handheld Garmin eTrex 20x GPS unit (Garmin Ltd, Olathe, KA, USA). After completing the final monitoring session, we strategically placed twenty funnel traps in areas where axolotls were most frequently observed. These traps were checked three times daily, every eight hours, over a continuous seven-day period.

On October 26, 2017, we randomly released eight axolotls—four males (ages ranging from 2.36 to 4.11 years) and four females (ages ranging from 2.19 to 5.61 years)—at two locations on the north and south sides of the study pond in LCO. From October 27 to December 7, alternating teams of three to four volunteers from a pool of 25, conducted 57 radio-tracking sessions from a small rowboat. We typically scheduled two, and occasionally three, monitoring sessions per day on Mondays, Wednesdays, and Fridays: the first from 9:00–12:00, and the subsequent session(s) from 17:00–21:00. We increased our monitoring efforts in the late afternoons and evenings based on findings from a pilot study, which suggested that axolotls are more active during these times (unpublished data from authors). However, from December 4–7— during the final days of this study, radio-tracking sessions were conducted every three hours around the clock to obtain more accurate measurements of distance traveled per hour.

On March 12, 2018, we introduced five male-female pairs of axolotls at five distinct sites within the study Chinampa in Xochimilco, ensuring each site was at least five meters apart. The females ranged in age from 1.39 to 3.98 years, and the males ranged from 1.39 to 4.39 years. From March 12 to April 20, a group of four volunteers from a pool of ten alternated camping at the Chinampa from Monday to Wednesday each week. They conducted radio-tracking of the axolotls on foot at approximately three-hour intervals during this period. However, starting from 18:00 hours on April 16, we increased the frequency of radio-tracking sessions to every hour, continuing until the early morning of April 20.

To document water temperature throughout the day and at the time of each telemetry reading, we placed three HOBO Pendant temperature data loggers (model UA-002–64; Onset Computer Corporation, Bourne, MA, USA) in the LCO study area and eight in the Xochimilco study area. All loggers were positioned at the same depth to ensure consistent measurements, although weather conditions may have influenced water levels. For the temperature analysis, we used the average temperature recorded by all HOBO loggers within each study area at the specific time and hour of each telemetry observation.

## Home range estimation

To estimate the home ranges of individual axolotls in LCO and Xochimilco, we employed the Minimum Convex Polygon (MCP) and enhanced it with the concave k-neighbor method. This advanced approach connects each observation point to its nearest neighbors, forming a boundary that more accurately reflects the actual areas axolotls frequent by excluding extensive, irrelevant territories typically included in traditional MCP estimates [53]. Given that axolotls are strictly aquatic and do not venture outside their water habitats, conventional MCP estimates could overestimate their home ranges.

We conducted a detailed spatial analysis using Kernel Density Estimation (KDE) to identify key habitat areas for individual axolotls and all grouped axolotls independently for both study sites. This method estimates the probability density function of a random variable, allowing us to pinpoint areas of high use from location data. We processed the telemetry data using the sf package [54] to ensure compatibility with spatial analysis tools. We converted it from a tabular format to a spatial features object (sf), enabling direct spatial operations on the dataset.

The KDE model was implemented using the adehabitatHR package in R [55], with the kernelUD function to compute the estimates. We used the reference bandwidth method for smoothing parameter selection. The analysis generated the 50% KDE isopleth, representing the core habitat area where individuals are most frequently found. This isopleth was extracted using the getverticeshr function, providing a spatial representation of the high-density areas for each individual axolotl.

## Data analyses

To compare the biological characteristics and home range sizes between LCO and Xochimilco axolotls, as well as between sexes within each study area, we employed a series of statistical tests. Independent t-tests were used for

normally distributed data (mass), while Mann-Whitney U tests were applied to non-normally distributed data (age, length, and home range estimates). Independent t-tests were also used to compare environmental conditions, specifically temperature, between the two study areas.

In addition to these comparisons, we used separate generalized linear mixed models (GLMMs) for each study area to examine the influence of biological (age, sex, mass, and length), temporal (time of day, and days since release), and environmental (temperature) factors on axolotl movement, specifically daily and hourly travel distances. Given that the distance data were right-skewed and not normally distributed in both study areas, we applied GLMMs using the glmmTMB package with a Gaussian family and a log link function for all models. We chose this distribution because it is well-suited for modeling continuous, positively skewed data [56]. Furthermore, to account for individual variability and repeated measurements, we included axolotl ID as a random effect in all models.

We assessed collinearity among explanatory variables using Variance Inflation Factors (VIF) to detect multicollinearity in our models. Including both length and mass resulted in VIF values exceeding seven, which indicates multicollinearity. Since mass estimations are more accurate and biologically relevant than length measurements, we excluded length from all analyses. This helped improve model performance and reduce VIF values to below two. We identified the best model for each response variable by comparing AICc values, with model selection performed using the dredge function from the MuMIn package in R [57]. We conducted all statistical analyses using R (v4.4.1) within RStudio 2024.04.2 "Chocolate Cosmos" [58].

### Distance per hour model

We calculated the hourly travel distance for each axolotl by measuring the distance between consecutive observation points, including those taken up to three hours apart in LCO and up to two hours apart in Xochimilco. For observations that were more or less than one hour apart, we standardized the travel distance to an hourly rate by dividing the observed distance by the time elapsed between points. We then averaged these standardized distances to obtain a consistent measure of hourly travel distance. The explanatory variables in the hourly travel distance models were sex, mass, age, time of day, temperature, and quadratic temperature. We scaled continuous variables to improve model fit.

### Distance per day model

To calculate the daily total distance traveled by each axolotl, we summed the distances between all consecutive observation points recorded within each 24-hour period. Data from the final days of monitoring in both study areas were excluded because the increased frequency of telemetry readings during this time inflated the estimated daily distances compared to earlier days. The explanatory variables included in the daily distance travel models were sex, mass, age, and number of days since release into the study area.

## Results

We recorded a total of 455 spatial locations in LCO (mean = 56.88 per axolotl; range = 56–57) and 1072 spatial locations in Xochimilco (mean = 107.2 per axolotl; range = 105–110). There were no significant differences in mass, length, or age between axolotls from LCO and Xochimilco, nor between males and females (S3 Table).

At the end of the studies, we successfully recaptured one axolotl in LCO and two in Xochimilco. On January 26, 2018, we recaptured a 2.5-year-old female axolotl (A01) in LCO. Initially, she weighed 63.86 g and measured 20.6 cm. Upon recapture, her weight had increased to 82.2 g and her length to 21.5 cm, reflecting a gain of 18.34 g. In Xochimilco, a 2.5-year-old male axolotl (C5), originally weighing 64.3 g, was recaptured on May 8, 2018, showing a 3-gram mass increase and a 3-mm growth in length. Additionally, on May 24, 2018, we recaptured a 2.5-year-old female axolotl (A08) with an initial mass of 85.3 g, although her recapture weight was not recorded. While these recaptures were successful, several other axolotls were detected but evaded capture. After the study concluded in Xochimilco, we observed a great

egret (*Ardea alba*) capturing an axolotl from the canal. Additionally, local chinamperos reported witnessing another axolotl being taken by a great egret, further confirming predation in the area.

### Home range

The total MCP home range for all eight axolotls in LCO averaged 2,747 m² (range: 548–4,404 m²) (Fig 2), while ten axolotls in Xochimilco showed a smaller mean of 382 m² (range: 175–697 m²) (Fig 3). This difference was statistically significant (p < 0.001), indicating that axolotls in LCO utilize a much larger area than those in Xochimilco. Similarly, the KDE 50% areas for the eight axolotls in LCO averaged 1,640 m² (range: 211–3,673 m²), compared to a smaller mean of 204 m² (range: 35–455 m²) for the ten axolotls in Xochimilco. This difference was also statistically significant (p = 0.001). We found no significant differences in MCP or KDE 50% core areas between sexes in either study area (LCO and Xochimilco), additionally, there were no significant correlations between MCP or KDE 50% sizes and age in either location (S2 Table).

### Distance per hour

The mean water temperature at LCO, based on readings corresponding to telemetry data collection times, was 15.92°C (range: 14.58–17.73; n = 222), compared to 16.66°C at Xochimilco (range: 14.9–18.73; n = 439). An independent t-test confirmed that LCO was significantly cooler than Xochimilco (p < 0.001). When considering all temperature data recorded by the HOBO devices, more extreme values were observed. The overall mean water temperature at LCO, using all recorded data, was 16.34°C (range: 12.11–30.76; n = 8,307), while at Xochimilco, it was 16.52°C (range: 12.01–37.71; n = 14,744).The best-fitting model for LCO revealed a significant quadratic relationship between distance traveled per hour and water temperature (p = 0.002), with movement increasing with temperature up to a peak, then decreasing at higher temperatures (Fig 4). In Xochimilco, we found a similar significant quadratic relationship (p < 0.001) that mirrored the pattern observed in LCO (Fig 4). In addition, the LCO model revealed a significant effect of sex on distance traveled per hour, with males covering less distance compared to females (p = 0.022). While the time of day also had a significant impact on movement (p < 0.001), indicating an increase in activity levels during the later hours of the day. Neither sex nor time of day were significant in Xochimilco. Furthermore, sex, mass, and age were not significant in either study area.

### Distance per day

We analyzed the daily distances traveled by axolotls at two different sites. In LCO, the mean distance traveled per day was 70.37 meters (range: 6.79–217.09 meters). In contrast, axolotls in Xochimilco traveled a mean distance of 46.55 meters per day (range: 2.39–135.35 meters). Axolotls in LCO traveled statistically larger distances than those in Xochimilco (U = 3958, p < 0.0001).

The distance analysis for LCO showed that the number of days since release had a significant negative effect on the daily distance traveled by axolotls (p = 0.038; Fig 5), suggesting that daily distance decreased over time. Additionally, sex had a significant effect (p = 0.024), with males traveling shorter daily distances than females, averaging 54.33 meters per day compared to 86.75 meters for females, consistent with the pattern observed for hourly distances. Neither mass nor age were significant. For the Xochimilco axolotls, the number of days since release did not have a significant effect on daily distance traveled. However, age had a significant negative effect (p = 0.003; Fig 6), indicating that older axolotls traveled shorter distances. Sex and mass had no significant effect.

## Discussion

Our study provides insights into the movement behaviors and home range sizes of captive-bred axolotls released in an artificial wetland (LCO) and a restored chinampa within their native habitat, Lake Xochimilco. Axolotls in LCO and Xochimilco were not randomly distributed but instead showed a preference for certain microhabitats, as evidenced by high densities of observations in certain areas and minimal to no observations in others (Figs 2 and 3). In Xochimilco, axolotls

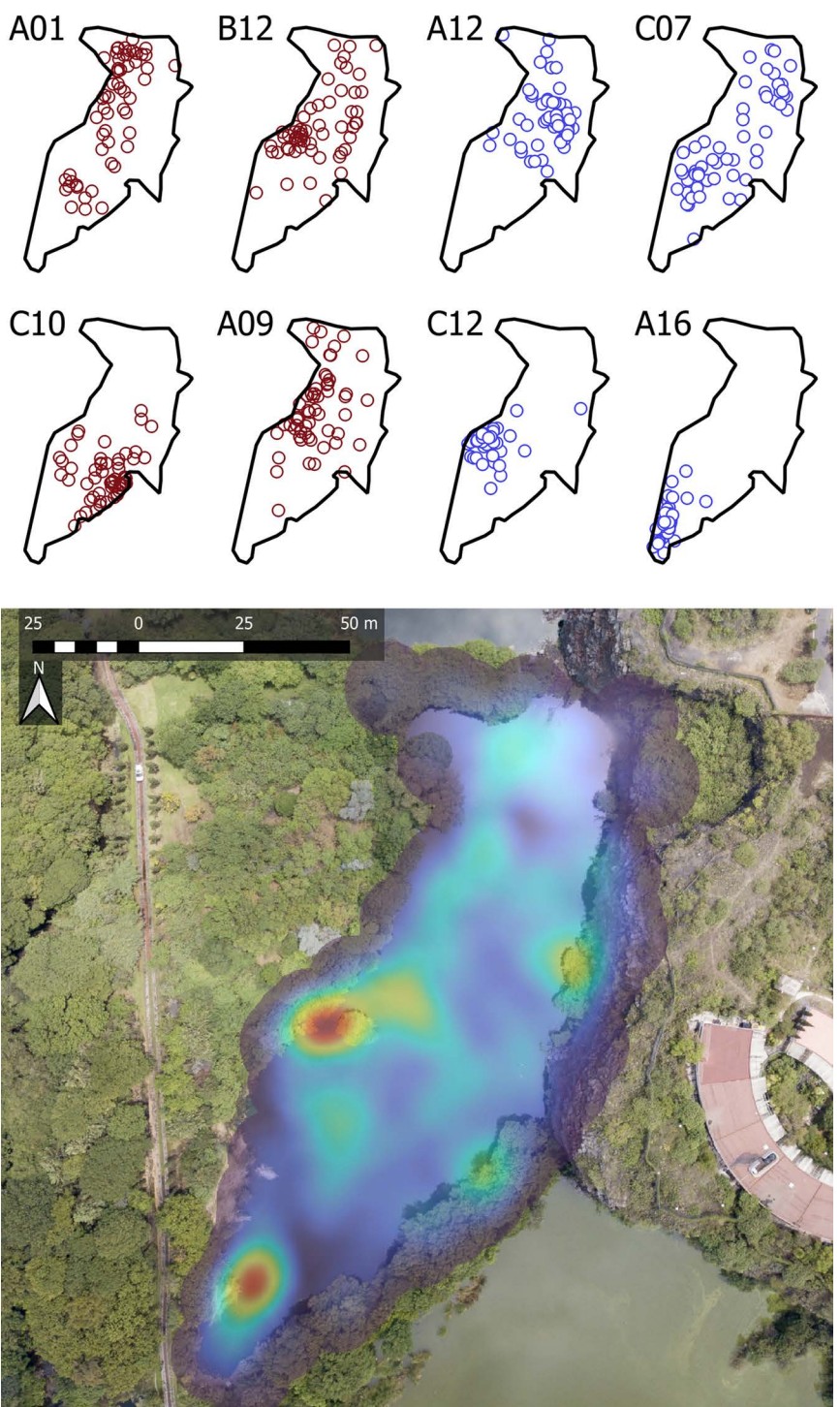

**Fig 2. Spatial distribution and habitat use in LCO.** The top panel illustrates the distribution of individual axolotls at La Cantera Oriente (LCO), separated by sex: males shown in blue (IDs: A12, C07, C12, A16) and females in red (IDs: A01, B12, C10, A09). The bottom panel features a Kernel Density Estimation (KDE) heatmap indicating habitat use intensity, with warmer colors denoting areas of higher use. The data for both panels are visualized on an aerial photograph taken by co-author David Schneider.

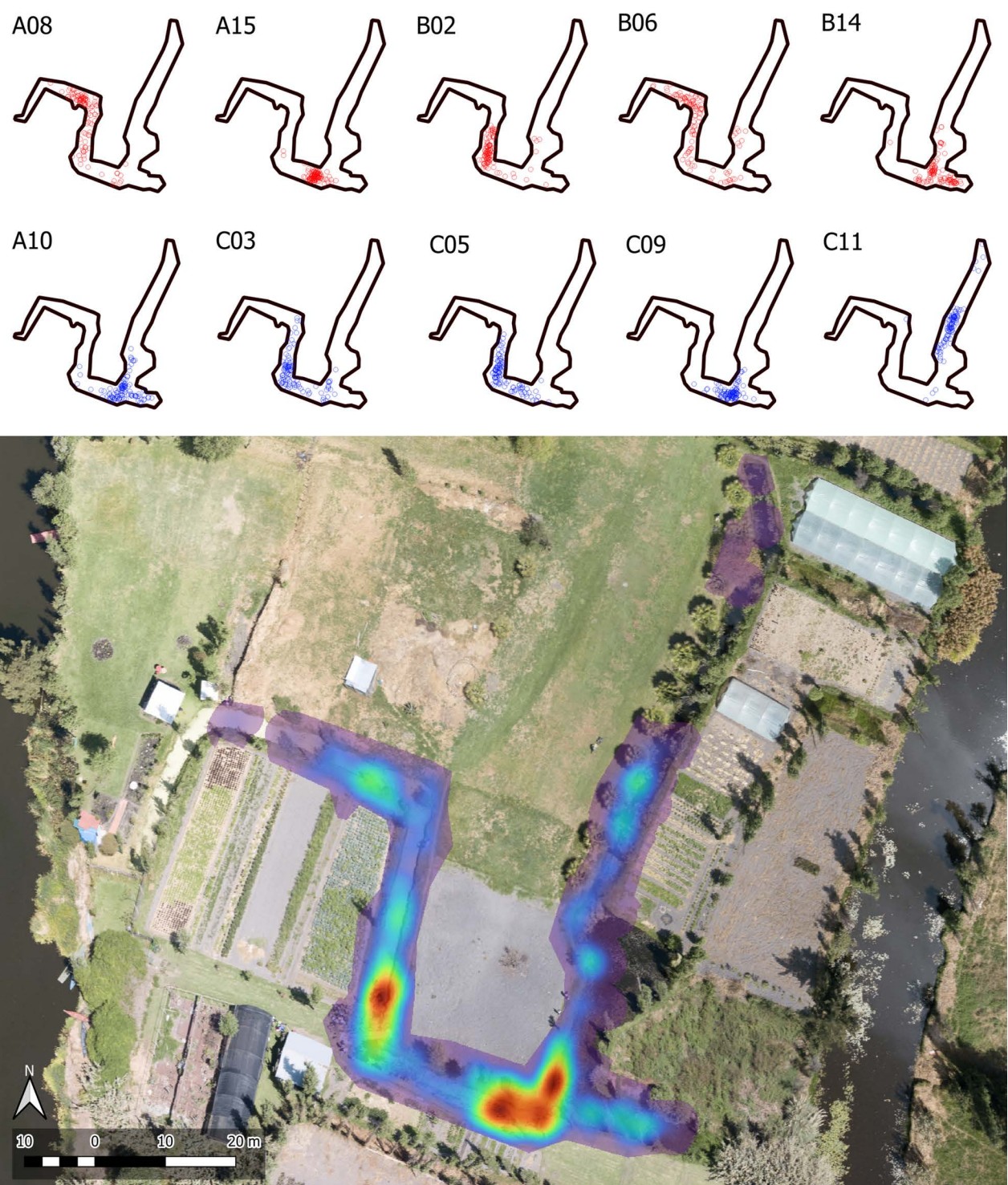

**Fig 3. Spatial distribution and habitat use in Xochimilco.** The top panel shows the distribution of individual axolotls in Xochimilco, differentiated by sex: males are depicted in blue (IDs: A10, C03, C05, C09, C11) and females in red (IDs: A08, A15, B02, B06, B14). The bottom panel displays a Kernel Density Estimation (KDE) heatmap that illustrates habitat use intensity, with warmer colors indicating areas of higher use. Both panels are visualized on an aerial photograph taken by co-author David Schneider.

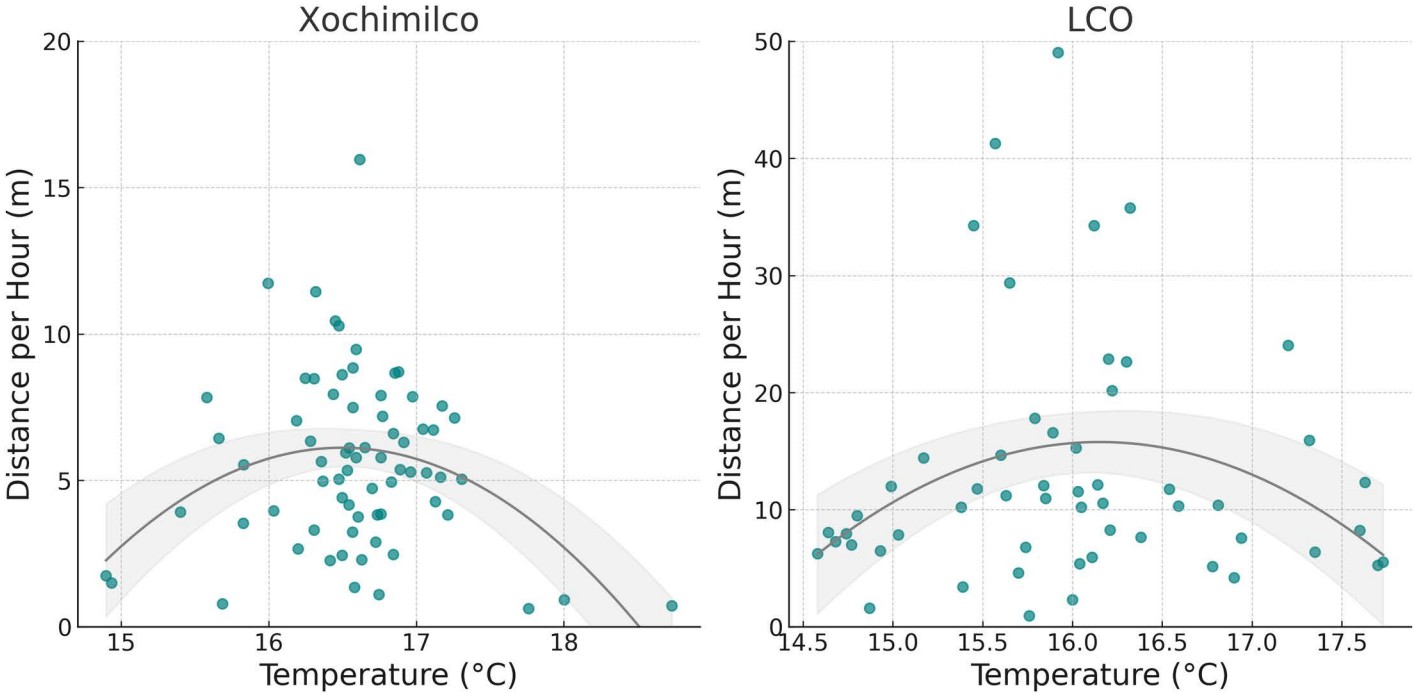

**Fig 4. Relationship between distance traveled per hour and water temperature.** The left panel shows data from Xochimilco, and the right panel shows data from LCO. Data points represent the average distance traveled per hour for each unique temperature value, plotted in teal. The grey line represents a quadratic trend line with corresponding confidence intervals.

notably avoided the westernmost section of the canal, where dense mats of floating duckweed (*Lemna gibba*) covered the water surface. Similarly, a study on spotted salamanders (*Ambystoma maculatum*) found fewer adults, egg masses, and larvae in pools with significantly more duckweed cover compared to those with less [59]. However, neither our study nor the other confirmed a causal relationship between duckweed presence and salamander distribution, the influence of duckweed on physicochemical parameters, such as dissolved oxygen and pH, warrants further investigation. Habitat preferences have been fairly documented in aquatic salamanders. Hellbenders (*Cryptobranchus alleganiensis*), for example, often hide under large, flat rocks and select microhabitats with gravel substrate over others [60,61]. Giant salamanders (*Andrias davidianus*) seek crevices that offer protection against predators while providing easy access to the surface [51]. Reintroduced axolotls (*A. mexicanum*), likewise, demonstrate a preference for aquatic vegetation, such as *Myriophyllum aquaticum* and *Eichhornia crassipes*, during daytime, likely as a strategy for predator avoidance during hours in which they are less active [47]. Additionally, these vegetated areas may serve as prey niches, acting as reservoirs for phytoplankton and zooplankton, which attract various vertebrates and invertebrates that function as prey for axolotls.

Xochimilco axolotls exhibited smaller home range sizes compared to those in LCO. This difference can likely be attributed to the considerably larger water area of the LCO pond, which is over 10 times the size of the Xochimilco canal. While we did not find direct evidence supporting this relationship in salamanders, water body size has been shown to be a critical factor in determining the home range sizes of freshwater fish [62,63]. Nevertheless, other factors might also influence movement patterns. For instance, research on hellbenders, another species of aquatic salamanders, suggests that habitat features, such as substrate and shelter availability, can significantly influence movement patterns and home range sizes [35,64]. In these studies, hellbenders released into areas with larger, more connected boulder formations tended to travel shorter distances, maintain more compact home ranges, and exhibit greater site fidelity compared to those placed

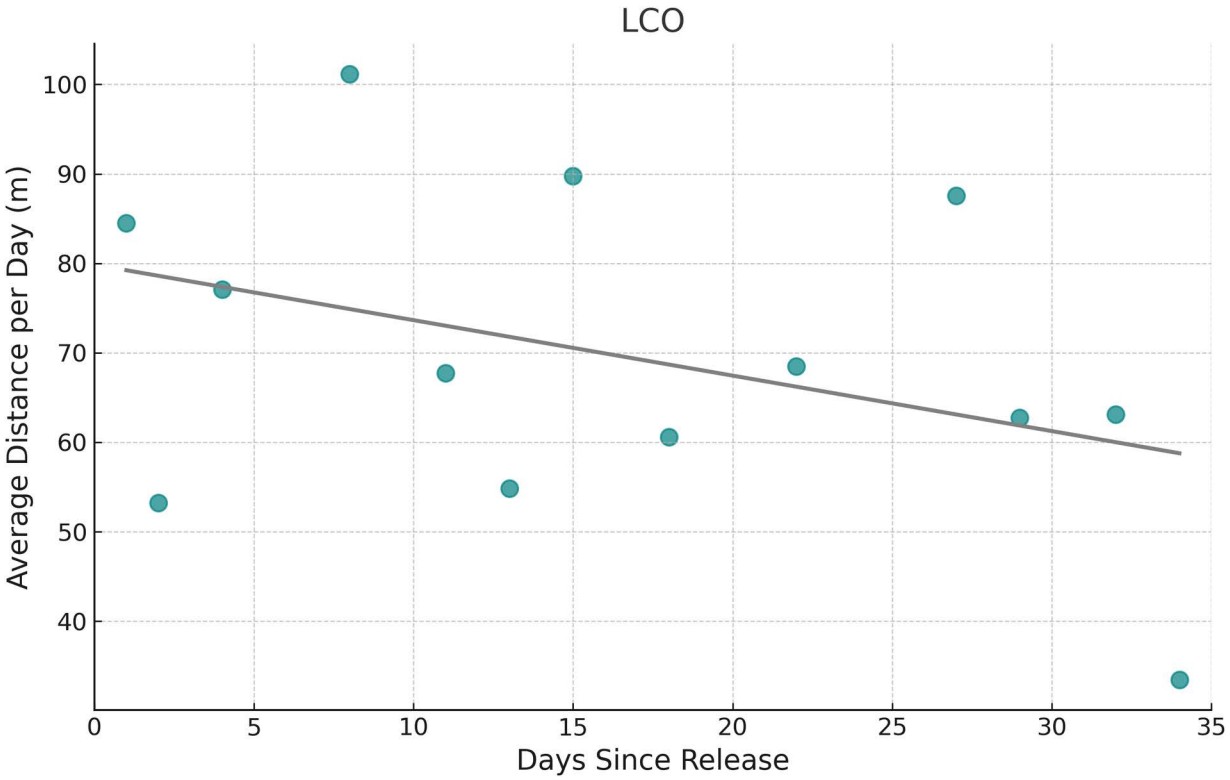

**Fig 5. Effect of time since release on daily travel distance in LCO axolotls.** Relationship between the number of days since release into the study area and the average distance traveled per day by axolotls.

in environments with more scattered boulders. Furthermore, another study found that translocated hellbenders moved greater distances and established larger home ranges compared to resident individuals, likely due to exploratory behavior, habitat suitability, and predation risks [65].

In LCO, axolotls showed a significant decrease in daily distance traveled as the number of days since release increased, suggesting that individuals reduced their movement over time as they became more settled in their environment. In contrast, axolotls in Xochimilco did not exhibit this pattern, likely because the smaller and simpler habitat provided quicker access to necessary resources. This finding from LCO is consistent with studies on hellbenders, which typically exhibit a short exploration phase before achieving stable settlement and high site fidelity within three weeks post-release. Similarly, captive-reared Chinese giant salamanders reintroduced to the wild were observed to settle in a single location within 10 days post-release [37]. Exploratory behavior is essential after translocations and reintroductions, as it allows animals to locate resources like food and shelter, while also assessing predator threats [66,67]. However, this behavior can also be costly, as it requires energy and heightens the risk of predation [42,68]. The larger, more complex LCO pond likely required axolotls to engage in a longer exploratory phase, as they navigated the environment to find optimal foraging and shelter sites, further contributing to their larger home ranges.

The distance traveled per hour by LCO axolotls showed a significant positive relationship with the time of day, with activity levels increasing in the afternoon and peaking around 9:00 pm. This rise in nighttime activity could be influenced by a combination of factors, including nocturnal foraging behavior, as observed in hellbenders [69], and predator avoidance, as documented in Northwestern salamanders (*Ambystoma gracile*) [70]. In a pilot study with two axolotls translocated into the LCO pond and monitored using VIF equipment, we observed that after an initial exploratory period of

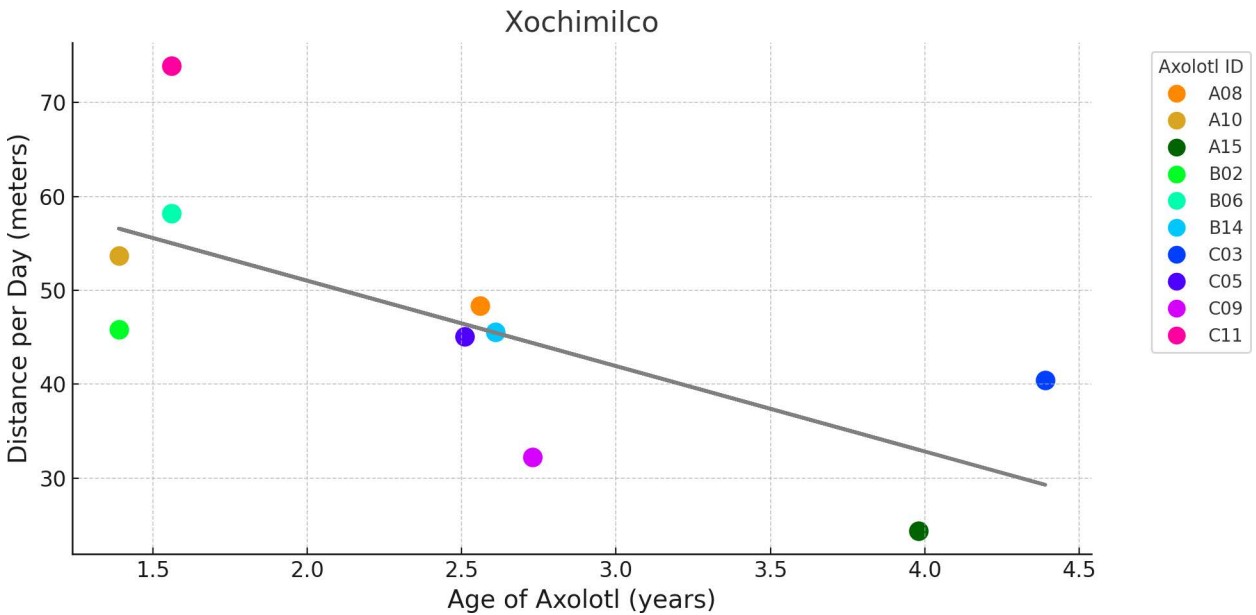

**Fig 6. Influence of age on daily travel distance in Xochimilco axolotls.** Each circle represents the average distance traveled per day by an individual axolotl, color-coded uniquely for each axolotl ID. Jitter was applied to distinguish overlapping points for axolotls B14 and C05.The grey line represents the linear trend of the data.

roughly one month, both individuals exhibited a clear shift in spatial use: they spent the daytime hours, when they were less active, in the northern part of the pond, and the nighttime hours in the southern section, where we suspect food availability was higher (unpublished data by the authors; S4 Fig). Although the current study did not reveal a distinct day-night spatial pattern, there were concentrations of daytime observations in northern areas, while nighttime movements were more frequent in the central and southern regions. In contrast, our current study in Xochimilco did not show a significant relationship between time of day and activity. However, in a previous experimental study conducted in Xochimilco, where we created distinct vegetated and non-vegetated quadrants, axolotls were more active at night and spent more time in non-vegetated areas [47]. The lack of a significant relationship in the current study could be due to the absence of environmental manipulation, leading to a more natural and homogenous distribution of vegetation compared to the experimental canal.

The relationship between axolotl movement, measured as distance traveled per hour, and temperature followed a nonlinear trend in both study areas. Axolotl movement increased with temperature up to a certain point, peaking at around 16–17°C in Xochimilco and 15.5–16.5°C in LCO and, beyond these optimal ranges, movement declined as temperatures continued to rise (Fig 6). This suggests that axolotls are most active within a narrow temperature window, and higher or lower temperatures may reduce movement, potentially reflecting physiological constraints. Previous research on amphibians has shown that temperature significantly influences their movement patterns, as they rely on external heat to regulate their metabolism and activity, particularly in response to environmental conditions [71]. Several studies have documented observed and preferred temperatures of salamanders in both field and laboratory conditions [72]. For example, *A. mexicanum* in laboratory settings selected temperatures between 16°C to 18°C [73], which resembles the optimal movement temperatures observed in our study. Similarly, mountain stream salamanders (*Ambystoma altamirani*) from Central Mexico were found at temperatures ranging from 16.42°C to 16.98°C [74]. However, several other *Ambystoma* species from Noth America have been associated with considerably higher temperatures, ranging from 24.8 °C to 34.6 °C [72].

On average, water temperatures in Xochimilco during March and April were significantly warmer than those in LCO from late October to early December. Interestingly, the minimum (12°C) and maximum (38°C) temperatures recorded in Xochimilco matched those reported in a single, shallow sample site from a 2003–2004 study [75]. However, our observed mean water temperature was lower, at 16.5°C, compared to the 20.5°C estimated from data across all sites in their study. It is important to note that our study did not include the warmest months of the year, whereas García et al.'s study collected data year-round. Given that we recorded identical extreme temperatures without sampling the warmest months, it is possible that maximum temperatures during the summer are now even higher, especially in shallow areas, potentially due to climate change's impact on rising temperatures. From a conservation perspective, this highlights the potential importance of LCO as a complementary long-term strategy for axolotl conservation, especially if it remains cooler than Xochimilco in the future. Additionally, the conditions at LCO may better reflect Lake Xochimilco's historical state, before the Aztecs transformed much of it into chinampas, a system of artificial islands used for agriculture that altered the natural water flow and ecosystem. Since axolotls evolved in the natural lake of pre-Aztec Xochimilco, they are likely better adapted to these historical conditions, which LCO may now better resemble.

At the individual level, our results show that age and sex significantly influenced axolotl movement measured by hourly and daily traveled distance. In Xochimilco, older axolotls traveled shorter daily distances compared to younger ones (Fig 5), while in LCO, males covered less hourly distance than females. However, we found no relationship between individual traits and home range sizes. While much research has focused on the influence of environmental factors on movement patterns like dispersal in amphibians, there remains limited empirical evidence regarding the role of individual traits, such as age, sex, and body size [36]. Juvenile Chinese giant salamanders have been observed to disperse farther than adults, likely moving away from their natal population [37], which aligns with our finding that younger axolotls were more mobile. Similarly, dispersal distance in red-backed salamanders (*Plethodon cinereus*) was significantly greater in juveniles than in adults [76], further supporting the idea that juveniles tend to be more mobile during early life stages across different species. This variation in movement patterns seen between juvenile and adult salamanders could also be attributed to territorial behavior in adults, as adults are known to establish and defend territories [37]. Though environmental factors may influence space use differently based on age and sex, identifying the sex of salamanders can be particularly challenging [77]. Additionally, studies that include age often categorize it into distinct life stages (larvae, juvenile, and adult), likely because they lack precise age estimates, rather than treating it as a continuous variable as we did in this study.

Our study demonstrates that captive-bred axolotls can survive and successfully forage in both their native habitat of Xochimilco and an artificial wetland environment like LCO. The observation of weight gain in recaptured individuals from both locations indicates effective foraging. However, predation risks remain a significant concern, particularly in Xochimilco, where we directly observed two predation events involving great egrets (*Ardea alba*) attacking adult axolotls. Great egrets are present in both study areas and are classified as migratory birds [78]. They are most abundant in Mexico City during the winter; however, observations of these birds can be made year-round (authors' observation). This highlights the vulnerability of axolotls to predation even at the adult stage. While we did not study fish presence, their potential impact on axolotl eggs as well as larval and juvenile stages through predation cannot be overlooked. Although invasive fish elimination efforts have been implemented in both study areas, continued monitoring and management are necessary to mitigate their ongoing threat.

One of the main causes of failure in reintroduction and translocation programs is the high mortality due to predation that occurs after release [79,80]. Animals born and raised in captivity do not learn to recognize and respond appropriately to predators and may completely lose their anti-predator behaviors, or these behaviors may become less efficient compared to those of wild animals [81]. Amphibians may be particularly at risk, as their small size makes them highly susceptible to predation by a wide range of predators, including reptiles, birds, and mammals [71].

The use of animal behavior as a tool for conservation has shown promise in improving survival rates of threatened species reintroduced into the wild [82]. 'Pre-release training' can play a crucial role in mitigating predation risks and boosting the success of reintroductions. Research on mammals, birds, and fish indicates that captive-born individuals can learn to recognize and respond to predators if given the appropriate training before release [81]. For example, Puerto Rican

parrots (*Amazona vittata*) exposed to cues simulating predation risk from hawks were less likely to be preyed upon by raptors compared to those that did not receive training [83]. Similarly, amphibian embryos and larvae can be trained with chemical stimuli to recognize predators [84]. Notably, responses to these stimuli may vary with age; for example, larvae of the giant American salamander (*Cryptobranchus alleganiensis*) trained to respond to trout cues remained still at around 4 months old but fled at around 6 months old [85].

Applying pre-release training to axolotls could reduce predation and enhance survival post-release. Temporary shelters could serve as pre-release enclosures where axolotls are exposed to visual and olfactory predator cues, mimicking real-life threats to help condition their anti-predator behaviors before they are released into the wild. In a previous study, shelters were successfully used in LCO where adult axolotls reproduced, and their eggs hatched into larvae, which then grew into juveniles [46]. This approach could enhance the success of reintroduction programs by offering axolotls a safer, semi-natural environment during the critical early stages of development, thereby strengthening their behavioral responses to threats and increasing their chances of long-term survival in the wild.

Our findings provide valuable insights into the movement ecology and habitat use of axolotls, informing conservation efforts aimed at their reintroduction and habitat management. By demonstrating the ability of captive-bred axolotls to survive and forage in both native and artificial environments, we reinforce the potential of artificial wetlands like LCO to serve as supplementary habitats for this critically endangered species. The elevated predation risks observed in Xochimilco highlight the necessity of implementing strategies such as pre-release predator training to enhance survival rates. Additionally, while our study did not directly compare temperatures simultaneously across sites, the observed differences suggest that LCO may offer a cooler, more stable environment. This could be particularly advantageous given the expected impacts of climate change on axolotl habitats. Most of Lake Xochimilco faces risks of agrochemical pollution due to agricultural activities but restored chinampas that avoid agrochemical use provide critical refuges and demonstrate sustainable conservation potential [86]. Continued monitoring and comparison of environmental variables such as temperature and water quality will be crucial in refining the use of LCO as a complementary conservation tool alongside Xochimilco. Future research should focus on refining pre-release training methods, assessing long-term survival and reproduction post-release, and comparing key environmental factors between LCO and Xochimilco to refine conservation strategies. Overall, our study contributes to a better understanding of axolotl ecology and offers practical approaches to improve conservation outcomes for this iconic species.

## Supporting information

**S1 Table. Summary statistics for physicochemical parameters measured in LCO and Xochimilco.** The table includes the mean, standard deviation (Std Dev), minimum (Min), and maximum (Max) values for pH, dissolved oxygen (DO, expressed as both percentage saturation and concentration in mg/L), conductivity (µS/cm), and salinity. Data were collected daily on monitoring days throughout the duration of the study.
(DOCX)

**S2 Table. Individual traits of LCO and Xochimilco axolotls.** Summary of individual characteristics of axolotls studied across two different study areas, LCO and Xochimilco. The table lists the identification code (ID), sex, age (in years), body mass (in grams), and length (in centimeters) of each axolotl.
(DOCX)

**S3 Table. Biological and spatial parameter comparison of axolotls by location and sex.** Comparison of individual traits (mass, length, and age) and home range (MCP and KDE 50%) between axolotls from LCO and Xochimilco, and between sexes within each location. Values are presented as means with ranges in parentheses. Significant p-values are highlighted in bold.
(DOCX)

**S4 Fig. Spatial distribution of axolotl observation points in LCO.** This figure displays the spatial distribution of axolotl observation points in La Cantera Oriente (LCO). The pilot study data are shown on the left, and the current study data on the right. Yellow points represent axolotl locations observed during the day, while purple points denote nighttime locations. This map was created using was created using OpenStreetMap data, which is licensed under the Open Database License (ODbL). (TIFF)

## Acknowledgments

We are grateful to the many field assistants and volunteers who contributed their time and effort to this study in the Xochimilco chinampa and La Cantera Oriente. We especially acknowledge Carlos Uriel Sumano Arias, Esmeralda Benitez Sandoval, Omar Jiménez, Cyntia Giselle Pensado Ortega, Araceli Mejía Barrio, Pedro Alberto Cabrera Castillo, Lourdes Itzel Murillo Reyes, Ana Yunuen Aguilera León, Allison Lory Bettencourt, Diana Laura Vázquez Mendoza, and Fabiola Jocelyn Real Reyes, who assisted in the field across one or both study areas. We also thank Pedro Méndez for generously allowing us to use his chinampa as a study area in Xochimilco.

## Author contributions

**Conceptualization:** Alejandra G. Ramos, Horacio Mena, Luis Zambrano.

**Formal analysis:** Alejandra G. Ramos, David Schneider.

**Funding acquisition:** Luis Zambrano.

**Investigation:** Alejandra G. Ramos.

**Methodology:** Alejandra G. Ramos.

**Project administration:** Alejandra G. Ramos, Horacio Mena, Luis Zambrano.

**Resources:** Horacio Mena, Luis Zambrano.

**Supervision:** Alejandra G. Ramos, Luis Zambrano.

**Visualization:** Alejandra G. Ramos, David Schneider.

**Writing – original draft:** Alejandra G. Ramos.

**Writing – review & editing:** Alejandra G. Ramos, Horacio Mena, David Schneider, Luis Zambrano.

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
