## [Editor Report · Decision Letter 0]

19 Nov 2024

PONE-D-24-51028Movement ecology of captive-bred axolotls in restored and artificial wetlands: conservation insights for amphibian reintroductions and translocationsPLOS ONE

Dear Dr. Zambrano,

Thank you for submitting your manuscript to PLOS ONE. After careful consideration, we feel that it has merit but does not fully meet PLOS ONE’s publication criteria as it currently stands. Therefore, we invite you to submit a revised version of the manuscript that addresses the points raised during the review process.

We look forward to receiving your revised manuscript.

Kind regards,

SSS Sarma

Academic Editor

PLOS ONE

Journal Requirements:

2. Thank you for stating the following financial disclosure: [This project was funded by UNAM PAPIIT No. 705 IV200117 and IV210117 Programa de Apoyo a Proyectos de Investigación e Innovación Tecnológica (PAPIIT-IV200117) AGR received a postdoctoral research grant from PAPIIT IV200117 and IV210117].

3. In the online submission form, you indicated that [Some data supporting this study are shared within the supplementary materials of this manuscript. However, sensitive data cannot be shared publicly at this time due to the endangered status of the axolotl and ongoing research. Requests for access to this data may be considered for conservation and research purposes and can be directed to the corresponding author at zambrano@ib.unam.mx ].

4. We note that Figure 1,2, 3 and S3 Fig in your submission contain [map/satellite] images which may be copyrighted. All PLOS content is published under the Creative Commons Attribution License (CC BY 4.0), which means that the manuscript, images, and Supporting Information files will be freely available online, and any third party is permitted to access, download, copy, distribute, and use these materials in any way, even commercially, with proper attribution. For these reasons, we cannot publish previously copyrighted maps or satellite images created using proprietary data, such as Google software (Google Maps, Street View, and Earth). For more information, see our copyright guidelines: http://journals.plos.org/plosone/s/licenses-and-copyright .

1. You may seek permission from the original copyright holder of 1,2, 3 and S3 to publish the content specifically under the CC BY 4.0 license.

We recommend that you contact the original copyright holder with the Content Permission Form (http://journals.plos.org/plosone/s/file?id=7c09/content-permission-form.pdf ) and the following text:

“I request permission for the open-access journal PLOS ONE to publish XXX under the Creative Commons Attribution License (CCAL) CC BY 4.0 (http://creativecommons.org/licenses/by/4.0/ ). Please be aware that this license allows unrestricted use and distribution, even commercially, by third parties. Please reply and provide explicit written permission to publish XXX under a CC BY license and complete the attached form.”

Additional Editor Comments:

Conservation of amphibians, particularly, paedomorphic salamanders is a problematic due to their low natural survival rates, reduced diet breadth, high sensitivity to stress.

Most laboratory-reared axolotl often fail to survive in nature because the conditions in which they are maintained are different from those in the field. The authors in this work have aimed at recreating the conditions in their natural habitat to other waterbodies. For this authors have radio-tracked the introduced individuals. The data presented here could form basis for future investigation in conservation biology.

However, the manuscript requires some corrections before it is sent out for experts’ comments.

The following may be useful to revise the manuscript:

1. Please provide full justification of presenting the field data of more than 7 years. Water quality variables then and now could be different. Even the final observations were terminated more than 4 years ago.

2. It appears that authors have re-captured at least some of the same individuals. Some kind of repeated anova or its equivalent may be needed.

3. Please provide a table of selected physico-chemical variables of water in La Cantera Oriente at the beginning of the experiments.

4. Authors did not observe any mortality of axolotl in the test lake. Do fish species occur in this lake? If so how competition and/or predation did not cause mortality of larval axolotl.

5. Some minor corrections:

Please provide authority of taxon Ambystoma mexicanum in the first mention

Line 167: artemia as Artemia

Lines 164-171 please use past tense for M&M section

Refs are too many.

---

## [Author Response · Author response to Decision Letter 1]

1 Dec 2024

JOURNAL REQUIREMENTS:

Thank you for your guidance. We have revised the manuscript to ensure it adheres to PLOS ONE's style requirements, including file naming, following the templates provided.

2. Thank you for stating the following financial disclosure: [This project was funded by UNAM PAPIIT No. 705 IV200117 and IV210117 Programa de Apoyo a Proyectos de Investigación e Innovación Tecnológica (PAPIIT-IV200117) AGR received a postdoctoral research grant from PAPIIT IV200117 and IV210117].

Thank you for your comment regarding the financial disclosure. The funders had no role in the study design, data collection and analysis, decision to publish, or preparation of the manuscript. We have included the suggested amended "Role of Funder" statement in the cover letter as requested and kindly ask for your assistance in updating the online submission form accordingly.

3. In the online submission form, you indicated that [Some data supporting this study are shared within the supplementary materials of this manuscript. However, sensitive data cannot be shared publicly at this time due to the endangered status of the axolotl and ongoing research. Requests for access to this data may be considered for conservation and research purposes and can be directed to the corresponding author at zambrano@ib.unam.mx].

Data are available on Figshare at https://doi.org/10.6084/m9.figshare.27936111 (inactive during the peer review process). During peer review, the dataset can be accessed via the private link: https://figshare.com/s/5aae1c742de87b2ad75f. The data will be made publicly available upon acceptance of the manuscript. We have updated the data availability statement in our revised manuscript (Lines 615–617).

4. We note that Figure 1,2, 3 and S3 Fig in your submission contain [map/satellite] images which may be copyrighted. All PLOS content is published under the Creative Commons Attribution License (CC BY 4.0), which means that the manuscript, images, and Supporting Information files will be freely available online, and any third party is permitted to access, download, copy, distribute, and use these materials in any way, even commercially, with proper attribution. For these reasons, we cannot publish previously copyrighted maps or satellite images created using proprietary data, such as Google software (Google Maps, Street View, and Earth). For more information, see our copyright guidelines: http://journals.plos.org/plosone/s/licenses-and-copyright.

Thank you for your detailed feedback concerning the licensing of the figures in our manuscript. We appreciate your guidance in ensuring that all visual content adheres to the Creative Commons Attribution License (CC BY 4.0). Below, we outline how we have addressed the concerns for each figure mentioned:

Figure 1: This figure displays a series of maps/images at various scale levels, illustrating the spatial location of two study areas. The broader-scale maps (country and Mexico City levels) utilize base map sources: Esri, Maxar, Earthstar Geographics, and the GIS User Community. The most detailed view, representing the LCO and Xochimilco study areas, now feature aerial photographs taken by co-author David Schneider, who has granted permission for publication under the CC BY 4.0 license.

Figure 2: This figure has been updated to include an aerial photograph captured by Dr. Schneider during the study. Permission has been granted by Dr. Schneider to publish the image under the CC BY 4.0 license.

Figure 3: We made no changes to this image since it is also an aerial image taken by Dr. Schneider.

Figure S3: This figure was created using R statistical software, incorporating data from OpenStreetMap, an open data platform. The base map data utilized is licensed under the Open Database License (ODbL) by the OpenStreetMap Foundation (OSMF). This license allows for the free use, distribution, transmission, and adaptation of the data by any party.

Please note that we have revised and updated all figure legends to include source and licensing information:

Figure 1. Spatial location of study areas. Overview of the two study areas in southern Mexico City: Xochimilco (XOC) and La Cantera Oriente (LCO). The top panels illustrate the geographic location of the study areas at three distinct spatial scales (base map sources: Esri, Maxar, Earthstar Geographics, and the GIS User Community), while the bottom panels provide close-up views of each wetland (aerial photographs taken by David Schneider). Scale bars represent distances in kilometers (top) and meters (bottom) for reference.

Figure 2. Spatial distribution and habitat use in LCO. The top panel illustrates the distribution of individual axolotls at La Cantera Oriente (LCO), separated by sex: males shown in blue (IDs: A12, C07, C12, A16) and females in red (IDs: A01, B12, C10, A09). The bottom panel features a Kernel Density Estimation (KDE) heatmap indicating habitat use intensity, with warmer colors denoting areas of higher use. The data for both panels are visualized on an aerial photograph taken by David Schneider.

Figure 3. Spatial distribution and habitat use in Xochimilco. The top panel shows the distribution of individual axolotls in Xochimilco, differentiated by sex: males are depicted in blue (IDs: A10, C03, C05, C09, C11) and females in red (IDs: A08, A15, B02, B06, B14). The bottom panel displays a Kernel Density Estimation (KDE) heatmap that illustrates habitat use intensity, with warmer colors indicating areas of higher use. Both panels are visualized on an aerial photograph taken by David Schneider.

S3 Fig. Spatial distribution of axolotl observation points in LCO. This figure displays the spatial distribution of axolotl observation points in La Cantera Oriente (LCO). The pilot study data are shown on the left, and the current study data on the right. Yellow points represent axolotl locations observed during the day, while purple points denote nighttime locations. This map was created using was created using OpenStreetMap data, which is licensed under the Open Database License (ODbL).

Finally, we have added the characteristics of the drone used by Dr. Schneider to capture the aerial photographs in the methods section of our manuscript (Lines 136 - 142). Additionally, we have attached a consent form authorizing the publication of these images under the CC BY 4.0 license.

ADDITIONAL EDITOR COMMENTS:

Conservation of amphibians, particularly, paedomorphic salamanders is a problematic due to their low natural survival rates, reduced diet breadth, high sensitivity to stress. Most laboratory-reared axolotl often fail to survive in nature because the conditions in which they are maintained are different from those in the field. The authors in this work have aimed at recreating the conditions in their natural habitat to other waterbodies. For this authors have radio-tracked the introduced individuals. The data presented here could form basis for future investigation in conservation biology.

However, the manuscript requires some corrections before it is sent out for experts’ comments.

The following may be useful to revise the manuscript:

1. Please provide full justification of presenting the field data of more than 7 years. Water quality variables then and now could be different. Even the final observations were terminated more than 4 years ago.

We acknowledge that the field data were collected more than seven years ago, and ideally, these results would have been published earlier. However, the timeline reflects the complexity and thoroughness of the larger, multi-step conservation project focused on axolotls, which includes separate studies addressing different aspects of conservation efforts including:

1. Telemetry experimental study in Xochimilco (Ayala et al. 2019), which focused on restoration efforts and experimental releases of axolotls in Xochimilco, was based on data collected in 2011. We finalized and published this work in 2019 while working on the current study, as it provided essential context for the conservation efforts presented here.

2. Pilot telemetry study in LCO, conducted in January 2017, involved releasing two axolotls in LCO to evaluate their survival and movement in this environment. Although we analyzed the data and prepared a draft manuscript for this study, we ultimately decided not to publish it to avoid redundancy with the current manuscript, which includes data from a larger sample of individuals. For transparency, we have included the draft of this study at the end of the responses to reviewers.

3. Temporary shelters study in LCO (Ramos et al. 2021), conducted from July to November 2017 in LCO, demonstrated the suitability of this site by confirming that axolotls could survive, reproduce, and complete their life cycle in one of the lakes. This study also documented key physicochemical variables, as well as the diversity and abundance of food resources available for axolotl larvae.

4. The current study integrates telemetry data collected from October to December 2017 in LCO and in 2018 in Xochimilco. It builds on the findings from these earlier studies and focuses on movement ecology in restored and artificial wetlands.

Publishing the earlier studies required significant time, as they were necessary to provide essential context and lay the groundwork for understanding axolotl habitat use and survival. This comprehensive foundation enables a deeper interpretation of the findings presented in the current manuscript. During this period, the lead author, Dr. Alejandra Ramos, transitioned to a new academic position at Universidad Autónoma de Baja California, taking on new teaching responsibilities, developing her independent research program, and addressing conservation priorities in Baja California. These professional transitions further contributed to the extended timeline for manuscript preparation.

Despite the timeline, we believe the ecological relevance of this study remains strong. It offers essential insights into amphibian reintroductions and translocations in both restored and artificial habitats. Moreover, this manuscript serves as an important foundation for future studies currently underway, including investigations into axolotl social behavior and temperature preferences. These studies build directly on the findings presented here, demonstrating the continued relevance of this dataset to ongoing conservation research.

Ayala, C., Ramos, A. G., Merlo, Á., & Zambrano, L. (2019). Microhabitat selection of axolotls, Ambystoma mexicanum, in artificial and natural aquatic systems. Hydrobiologia, 828(1), 11-20.

Ramos, A. G., Mena‐González, H., & Zambrano, L. (2021). The potential of temporary shelters to increase survival of the endangered Mexican axolotl. Aquatic Conservation: Marine and Freshwater Ecosystems, 31(6), 1535-1542.

2. It appears that authors have re-captured at least some of the same individuals. Some kind of repeated anova or its equivalent may be needed.

You are correct that our movement analyses (daily and hourly travel distances) involved repeated measurements from the same individuals. To address this, we used generalized linear mixed models (GLMMs) with axolotl ID as a random effect, which accounts for the non-independence of repeated measurements within individuals. To clarify this in the manuscript, we revised the sentence “Axolotl ID was incorporated as a random effect in all models to account for individual variability” to “To account for individual variability and repeated measurements, axolotl ID was included as a random effect in all models” (Line 284).

We have carefully reviewed our other analyses, which involved t-tests for comparisons, and can confirm that these analyses were based on independent data points. The comparisons of biological characteristics (mass, age, and length) utilized single measurements taken from each individual, ensuring no repeated data. Similarly, the home range comparisons (MCP and KDE 50% areas) were based on aggregated metrics that summarize each individual’s spatial use over the entire study period. As each individual contributed only one value to these analyses, the data were independent, and there was no need to control for repeated measurements.

3. Please provide a table of selected physico-chemical variables of water in La Cantera Oriente at the beginning of the experiments.

Thank you for your suggestion. To address this, we have included a table in the supplementary materials (Table S1) summarizing the selected physico-chemical parameters of the water recorded throughout the study for both La Cantera Oriente (LCO) and Xochimilco. The table includes variables such as temperature, dissolved oxygen, pH, and conductivity, providing a comprehensive overview of the environmental conditions across the duration of the study. We have also updated the Materials and Methods section to reflect this addition and ensure that readers can locate the information easily (Lines 142–144).

4. Authors did not observe any mortality of axolotl in the test lake. Do fish species occur in this lake? If so how competition and/or predation did not cause mortality of larval axolotl.

Thank you for raising this important concern about potential competition or predation in the test lake. We would like to clarify that our study focused exclusively on adult axolotls. We did not study or monitor larval stages, and therefore, mortality related to larval axolotls is beyond the scope of this research.

However, we agree that this is a valuable point. Axolotls, particularly larvae, would likely experience fish predation in both study areas, and this could be an important factor influencing their survival. We have revised the discussion section to address the potential impact of fish predation on eggs, larvae, and juvenile axolotls. While we did not specifically study fish presence or its influence, the updated text acknowledges their potential threat and emphasizes the importance of continued efforts to eliminate invasive fish species and monitor their presence in both study areas (Lines 553 – 557).

Some minor corrections:

Line 167: artemia as Artemia. Done, thank you.

Lines 164-171 please use past tense for M&M section.

Thank you for your observation regarding the tense used in the Materials and Methods section. The subsection in question details the husbandry practices at the LRE colony, which were established prior to our study and continue to the present day. To emphasize the ongoing nature of these practices and ensure clarity, we have maintained the use of present tense. However, to avoid confusion, we have now clarified in the manuscript that these conditions have been consistently in place from before the study began and are still in effect. We hope this modification helps eliminate any confusion regarding the temporal application of these practices:

“The housing conditions in this colony, which have been consistently implemented since bef

---

## [Decision Letter · Decision Letter 1]

4 Feb 2025

PONE-D-24-51028R1Movement ecology of captive-bred axolotls in restored and artificial wetlands: conservation insights for amphibian reintroductions and translocationsPLOS ONE

Dear Dr. Zambrano,

Thank you for submitting your manuscript to PLOS ONE. After careful consideration, we feel that it has merit but does not fully meet PLOS ONE’s publication criteria as it currently stands. Therefore, we invite you to submit a revised version of the manuscript that addresses the points raised during the review process.

We look forward to receiving your revised manuscript.

Kind regards,

Academic Editor

PLOS ONE

Additional Editor Comments:

Dear Authors:

The reviewer one could not add detailed comments to his/her report. Please consider the following too:

Line 27.- I do not believe that the concept of the Mexican axolotl is well applied since there are more species of axolotls in Mexican territory, so the best concept is the Xochimilco axolotl.

Line 28.- specify Xochimilco lake since putting only Xochimilco refers to the mayor's office or municipality.

Line 36.- specify values of how much they traveled or on average with the males

Line 54.- update quote speaks in the present tense and it's been 5 years, they suggest that half of these species are probably also threatened so I think the statistic has already changed.

Line 56-58. Habitat loss, fragmentation and degradation are among the main drivers of extinctions and declines in amphibian populations worldwide, but pollution of ecosystems is missing.

Line 62. Refer to Ambystoma mexicanum as the Xochimilco axolotl or Axolotl, since there are more species of axolotls in Mexico

Line 70-73. Emphasize current problems, it is read that this happened in pre-Hispanic times

Line 73-74. Who is this project from, who participates, what institutions?

Line 76. Why is the water quality so poor?

Line 80. Emphasize that translocation is one of those continuous and proactive human interventions to preserve vulnerable species and prevent extinctions

Line 91-92. Explain the importance of what you mention because the habitat of Ambystoma mexicanum is wetlands (Lake Xochimilco)

Line 105-106. We all know that the conservation of endangered species within their natural habitats remains the ideal strategy, but why?

Line 106. Mention Lake Xochimilco, not just Xochimilco because you are referring to the aquatic system, not the mayor's office, please check this throughout the document

Line 109. Justify why you mention the eastern quarry, you only mention it suddenly, can you mention, LIKE THE CASE OF……, AN AREA………

Line 117-119. You mention that given these characteristics, LCO could function as a potential habitat for axolotls, but you mention that it has rivers, lakes, fauna and flora, but what makes it ideal, for example temperature, water pH, is it not waste water, does it have invasive species?, the conductivity is perfect, the oxygen levels are similar to the demand of the organisms etc etc, why is it an alternative site for conservation efforts outside of its native environment?

Line 121- 131. I consider the physicochemical factors, which I do not see described in this paragraph or analyzed for a study of this type, as they are crucial.

Line 136. Place Lake Xochimilco

Line 144. Place Lake Xochimilco

Line 144. Indicate in the Cantera Oriente that it is a lake, lagoon, pool, etc. etc.

Line 151. In Lake Xochimilco we use ……

Line 155. Similar to those of the old Lake Xochimilco……..

Line 174. Ages of the axolotls

Line 213. Age of the axolotls?

Line 224. Age of the axolotls?

Line 226-230. This information is not relevant

Line 237. Place Lake Xochimilco

Line 244. Indicate which aquatic system is in (artificial wetland) LCO and in (lake) Xochimilco.

Line 167-169. Does the age of the axolotls have an influence?

Why didn't they take into account the age data, wouldn't they be important in their study, or why did they choose the age of the axolotls? Does the age influence the displacement?

Line 300-302. Up to this point the variable “age” is touched upon

Line 323. How many days after the release was this organism recaptured on May 24, what year?

Line 326-328. These comments should be expressed in the discussion

Line 332. Indicate Lake Xochimilco

Line 334. Indicate Artificial wetland

Line 415-417. You mention predation by herons on your organisms, however you do not discuss that this area of Mexico City is a place where birds migrate mainly in the winter, which would be a limitation for the presence of birds.

Line 418-427. Although your study and that of Ambystoma maculatum do not mention a relationship with the presence of Lemna gibba, you have to look for that relationship with this factor, it could be the low amount of dissolved oxygen in the water, the pH, etc. etc. physicochemical parameters that can be altered by the presence of plant cover.

Line 431-434. Discuss that these areas, in addition to being a refuge from predation, are a prey niche for axolotls, since they function as reservoirs of phytoplankton and zooplankton, attracting various groups of vertebrates and invertebrates that function as prey for axolotls.

Line 556-562. And what can you say about the pollutants in the water? It is known that the Xochimilco Lake area is an agricultural zone, and they use agrochemicals that are dumped into the canals by leaching, affecting the survival of amphibians that have permeable skin.

Sincerely

Handling Editor

Reviewers' comments:

Reviewer's Responses to Questions

**Comments to the Author**

1. If the authors have adequately addressed your comments raised in a previous round of review and you feel that this manuscript is now acceptable for publication, you may indicate that here to bypass the “Comments to the Author” section, enter your conflict of interest statement in the “Confidential to Editor” section, and submit your "Accept" recommendation.

Reviewer #1: (No Response)

Reviewer #2: All comments have been addressed

2. Is the manuscript technically sound, and do the data support the conclusions?

Reviewer #1: Partly

Reviewer #2: Yes

3. Has the statistical analysis been performed appropriately and rigorously? 

Reviewer #1: Yes

Reviewer #2: Yes

4. Have the authors made all data underlying the findings in their manuscript fully available?

Reviewer #1: Yes

Reviewer #2: Yes

5. Is the manuscript presented in an intelligible fashion and written in standard English?

Reviewer #1: Yes

Reviewer #2: Yes

6. Review Comments to the Author

Reviewer #1: (No Response)

Reviewer #2: In general, the ms regarding the movement ecology of the axolotls in both restored and artificial wetlands is of great relevance due to its current status and evidently the ecological importance. Additionally, authors have adequately addressed comments raised in a previous round of review.

In the abstract authors should mention some quantitative data, not only trends.

Line 99-101. What about the disadvantages? Limited range, weather conditions, organisms behavior etc. elaborate on the disadvantages and advantages of implementing this tool.

L415-417 Information has been previously reported in the ms.

7. PLOS authors have the option to publish the peer review history of their article (what does this mean? ). If published, this will include your full peer review and any attached files.

**Do you want your identity to be public for this peer review?** For information about this choice, including consent withdrawal, please see our Privacy Policy .

Reviewer #1: No

Reviewer #2: **Yes: ** Brenda Karen González Pérez

---

## [Author Response · Author response to Decision Letter 2]

10 Feb 2025

We attached a document with all our responses to the reviewers and editor titled "Response to Reviewers".

---

## [Decision Letter · Decision Letter 2]

19 Mar 2025

Movement ecology of captive-bred axolotls in restored and artificial wetlands: conservation insights for amphibian reintroductions and translocations

PONE-D-24-51028R2

Dear Dr. Zambrano,

We’re pleased to inform you that your manuscript has been judged scientifically suitable for publication and will be formally accepted for publication once it meets all outstanding technical requirements.

Kind regards,

SSS Sarma

Academic Editor

PLOS ONE

Additional Editor Comments (optional):

Reviewers' comments:

Reviewer's Responses to Questions

**Comments to the Author**

1. If the authors have adequately addressed your comments raised in a previous round of review and you feel that this manuscript is now acceptable for publication, you may indicate that here to bypass the “Comments to the Author” section, enter your conflict of interest statement in the “Confidential to Editor” section, and submit your "Accept" recommendation.

Reviewer #1: All comments have been addressed

Reviewer #2: All comments have been addressed

2. Is the manuscript technically sound, and do the data support the conclusions?

Reviewer #1: Yes

Reviewer #2: Yes

3. Has the statistical analysis been performed appropriately and rigorously? 

Reviewer #1: Yes

Reviewer #2: Yes

4. Have the authors made all data underlying the findings in their manuscript fully available?

Reviewer #1: Yes

Reviewer #2: Yes

5. Is the manuscript presented in an intelligible fashion and written in standard English?

Reviewer #1: Yes

Reviewer #2: Yes

6. Review Comments to the Author

Reviewer #1: I have noticed that you have taken into account the observations I previously made, so I believe that you are improving according to my observations.

Reviewer #2: All comments have been adressed or have been further explained or discussed. Therefore I suggest accepting the manuscript.

7. PLOS authors have the option to publish the peer review history of their article (what does this mean? ). If published, this will include your full peer review and any attached files.

**Do you want your identity to be public for this peer review?** For information about this choice, including consent withdrawal, please see our Privacy Policy .

Reviewer #1: No

Reviewer #2: **Yes: ** Brenda Karen Gonzalez Perez

---

## [Editor Report · Acceptance letter]

PONE-D-24-51028R2

PLOS ONE

Dear Dr. Zambrano,

I'm pleased to inform you that your manuscript has been deemed suitable for publication in PLOS ONE. Congratulations! Your manuscript is now being handed over to our production team.

Kind regards,

on behalf of

Professor SSS Sarma

Academic Editor

PLOS ONE